# Evaluating the Evolving Treatment Landscape of Systemic Therapies in Penile Cancer

**DOI:** 10.3390/cancers17182956

**Published:** 2025-09-10

**Authors:** Salvador Jaime-Casas, Regina Barragan-Carrillo, Federico Eskenazi, Juan P. Dugarte, Jad Chahoud, Philippe E. Spiess, Luis G. Medina

**Affiliations:** 1City of Hope Comprehensive Cancer Center, Department of Medical Oncology & Therapeutics Research, Duarte, CA 91010, USA; sjaimecasas@coh.org; 2Instituto Nacional de Cancerologia, Mexico City 14080, Mexico; rbarraganc@incan.edu.mx; 3USC Institute of Urology, Keck School of Medicine, University of Southern California, Los Angeles, CA 90033, USA; federico.eskenazi@med.usc.edu (F.E.); juan.dugarte@med.usc.edu (J.P.D.); 4Department of Genitourinary Oncology, H. Lee Moffitt Cancer Center and Research Institute, Tampa, FL 33612, USA; jad.chahoud@moffitt.org (J.C.); philippe.spiess@moffitt.org (P.E.S.); 5Department of Urology, Medical University of South Carolina, Charleston, SC 29425, USA

**Keywords:** penile cancer, squamous cell carcinoma, systemic therapy, clinical trials

## Abstract

Penile cancer is among the rarest genitourinary malignancies, yet it is also one of the most aggressive, with limited treatment options available that can preserve sexual and reproductive function. For patients with locally advanced or metastatic disease, systemic treatment has historically relied on cisplatin-based chemotherapy regimens. However, its limited efficacy and suboptimal long-term outcomes have underscored the urgent need for new therapeutic strategies. Ongoing clinical trials are now exploring the role of immunotherapy, immune checkpoint inhibitors, and other novel forms of targeted therapy. In parallel, advances in tumor microenvironment characterization, HPV-associated signatures, and molecular profiling are generating insights that may help identify biomarkers and actionable targets for future drug development. In this review, we summarize the current evidence for systemic therapy in advanced penile cancer, highlight ongoing trials, discuss potential biomarkers that may guide treatment selection, and outline emerging research in the molecular space. We aim to provide an up-to-date overview of the evolving treatment landscape to support clinicians and researchers in improving outcomes for patients with penile cancer.

## 1. Introduction

Penile cancer is a rare disease in developed countries, with an estimated prevalence of about 2000 men affected in the United States in 2025 [1]. Its incidence is higher in parts of Asia, Africa, and South America, where limited disease awareness and restricted healthcare access, especially among rural, low-income, and homeless populations, contribute to a delayed diagnosis [2,3]. The majority of penile cancer cases (95%) arise from the squamous cells of the glandular or preputial skin and are penile squamous cell carcinomas (PSCC) [4]. Tumoral lesions occur more commonly in the glans (35–48%), inner prepuce (13–21%), and penile shaft (10–15%) [5]. Other tumor histologies, such as melanoma, sarcoma, and Paget’s disease, can also occur on the penile and glandular skin [5]. The World Health Organization (WHO) classifies penile cancers based on their association with human papillomavirus (HPV) infection into HPV-related, accounting for approximately 40% of cases, and HPV-unrelated, representing the remaining 60% [6]. Common risk factors include smoking, poor hygiene, and HPV infection. Nodal involvement is the strongest prognostic factor as penile cancer spreads via lymphatic vessels, initially to the inguinal nodes and subsequently to the pelvic nodes [7]. Distant metastasis occurs in less than 5% of cases, and potential invasion sites include the lungs, liver, bone, and brain [8]. The 5-year cancer-specific survival (CSS) rate decreases with advancing stage, from 92% in pN0 tumors to 73% in N1, 61% in N2, 33% in N3, and 9% in M1 disease [9,10,11]. Unfortunately, up to 50% of patients present with locally advanced disease, and 1–10% harbor distant metastases [10,11]. In a single-center experience from Sweden, up to 65% of patients had a delay of more than 6 months in their initial diagnosis [12]. A significant number of patients will delay seeking medical attention, as lesions are usually painless. This delay can also be associated with psychosocial components like embarrassment, fear, and denial by the patient [13]. An early diagnosis and referral are paramount for optimizing outcomes, as lymphatic spread is associated with poor prognosis.

The treatment of localized penile cancer includes surgical resection, which often leads to anatomical disfigurement and poor functional outcomes [14]. Conversely, the cornerstone for metastatic disease is cisplatin-based systemic therapy. However, the response rate remains suboptimal [15,16,17]. Management becomes even more challenging for those with residual disease following surgical consolidation [18,19]. There is limited evidence to conclude that adjuvant chemotherapy could help these patients. Since overall survival (OS) is poor, its use should be focused on a patient-centered approach, offering palliative care to selected patients [20]. Several clinical trials, such as InPACT, AFU-GETUG 25, HERCULES, NCT04475016, and NCT05526989, reflect the ongoing efforts to improve the management and overall survival for patients with penile cancer. Evidence from these studies may potentially clarify the current understanding of the use of neoadjuvant and adjuvant radio-chemotherapy and the combination of chemotherapy with immunotherapy and tyrosine kinase inhibitors. This comprehensive review focuses on the current evidence of systemic therapy application in this disease while also exploring details of emerging therapeutic alternatives and ongoing clinical trials that could inform penile cancer management. We incorporate the most recent trial data available from recent years, including results from ongoing phase II and III studies. Our discussion will be organized around the design of therapeutic approaches, ranging from chemotherapy to immunotherapy, tyrosine kinase inhibitors, and vaccine-based strategies, creating a structure that follows the evolution of treatment options in the landscape of penile cancer.

## 2. Treatment Landscape for Penile Cancer

The specific therapeutic approach for managing PSCC is dictated by the degree of tumoral invasion. Locally invasive disease (cT0-2) can be managed surgically, with often organ-sparing approaches. Conversely, partial or radical penectomy leads to anatomical disfigurement and loss of function and is indicated in higher-grade disease with tumors extending to the corpus cavernosum or adjacent organs (cT3-T4). Due to the avidity of PSCC to show lymphatic invasion, inguinal and pelvic node dissections can be therapeutic in micrometastatic disease and diagnostic, as lymph node involvement would determine the need for systemic therapy [21]. In the following sections, we present the treatment landscape in an order that reflects the natural progression of disease, beginning with less aggressive stages and advancing to metastatic disease. This structure allows for a focused discussion with particular emphasis on advanced and metastatic presentations, where systemic therapy plays its most critical role in influencing patient outcomes.

## 3. Non-Palpable Inguinal Lymph Node Disease

Non-palpable inguinal lymph node disease can be broadly categorized as low (intraepithelial neoplasia, Ta, or T1) and intermediate/high-risk (T1b, or ≥T2). Low-risk disease can be safely managed with surveillance, which includes clinical evaluation every six months in the first two years and every 12 months for four years [22]. Thereafter, evaluation can be performed as clinically indicated. Up to 25% of patients with clinically node-negative disease (cN0) will harbor micrometastasis at diagnosis, urging risk-stratification in this patient population [22]. Notably, histopathological analysis will yield the most information for clinical decision-making. Higher tumoral grade and lymphovascular invasion can increase the risk of micrometastatic spread up to 50% [23,24]. For this reason, performing early lymphadenectomy or dynamic sentinel node biopsy (DSNB) in the context of high-risk features offers increased long-term survival compared to delayed lymphadenectomy, with patient survival over 90% with early lymphadenectomy and less than 40% with lymphadenectomy for patients with regional recurrence [25]. DSNB has a more established role for patients with clinically impalpable inguinal lymph nodes (cN0). Evidence from a systematic review encompassing 28 studies and 2893 patients showed that DSNB carries a relatively high specificity (0.82) for detecting positive lymph nodes. However, it remains a poor technique to discriminate which patients will show further metastatic involvement after completion of the ILND, suggesting that improved stratification techniques beyond DSNB might help avoid overtreatment and comorbidities [26]. Consequently, surveillance should only be offered to patients with low-risk characteristics, such as pTis/pTa tumors [27]. Long-term compliance should also be considered in such patients, as losing patients to follow-up can result in missing clinically significant disease. A high level of patient engagement is expected for this follow-up period, and clinicians should weigh the benefit of surveillance against the risk of losing patients to follow-up. Research is currently focused on determining molecular surrogates that could identify minimal residual disease to better select patients for surgical intervention and lymph node dissection. A retrospective analysis of 7 patients with advanced PSCC showed that ctDNA assays were useful for detecting minimal residual disease, as well as highly predictive of treatment response and imaging-based progression. However, this should be interpreted in the context of a small sample size that limits the generalizability of the results [28]. A similar retrospective of 13 patients with primary urethral cancer or PSCC evaluated 66 ctDNA tests to assess residual disease. ctDNA results showed a concordance rate of 98.5% based on radiographic disease status and/or surgical downstaging, with positive ctDNA tests showing an odds ratio (OR) for progression of 36.7 (4.5–299.5, *p* < 0.0001), and negative ctDNA OR for progression was 0.02 (0.003–0.2, *p* < 0.0001). Although both studies support the utility of ctDNA as a surrogate for response, prospective trials with larger sample sizes are warranted to validate these results [29].

## 4. Palpable Inguinal Lymph Node Disease

Clinically palpable lymphadenopathy (cN1-3) represents disease with overt regional lymphatic invasion. A thorough physical examination should be performed in all patients, including describing the character, dimensions, laterality, and mobility of the lymphadenopathy. Although it would not alter surgical management, imaging evaluation with CT or MRI is warranted to rule out distant metastatic spread. In select cases when clinical evaluation is dubious, image-guided fine needle aspiration with cytologic analysis and/or a core biopsy can help define staging and treatment [30]. According to the National Comprehensive Cancer Network (NCCN) and European Association of Urology (EAU) guidelines, neoadjuvant TIP-based chemotherapy (paclitaxel, ifosfamide, cisplatin) may be considered in cases of palpable or bilateral inguinal lymph node disease. However, these recommendations are based on retrospective evidence and have not been extensively validated. In select cases, TIP-based neoadjuvant chemotherapy followed by consolidative radical surgery and inguinal lymph node dissection can improve overall survival outcomes in up to 37% of patients [7,27,31]. However, in patients with non-bulky N1-N2 disease, the clinical benefit of neoadjuvant chemotherapy is even less established, and most recommendations are extrapolated from studies including patients with more advanced disease [31].

For patients with pN2-3 disease, pelvic lymph node dissection is warranted. If pelvic nodes are considered positive, adjuvant radiotherapy with or without chemotherapy is recommended [22]. Currently, there is insufficient data to inform the use of adjuvant chemotherapy. However, extrapolation from neoadjuvant data suggests that giving four courses of TIP in the adjuvant setting is reasonable if it was not given preoperatively and the lymph node dissection pathology shows high-risk features. 5-FU plus cisplatin can be considered as an alternative to TIP in the adjuvant setting, albeit data demonstrating its efficacy is limited [22].

### Neoadjuvant and Adjuvant Treatment

In patients with locally advanced disease (any T grade, N1-3), particularly those with palpable inguinal lymph node involvement, the use of perioperative chemotherapy may be supported by retrospective studies [32]. For example, a study evaluating 20 patients with penile cancer and bulky inguinal lymph node disease found that early lymph node dissection followed by adjuvant chemotherapy was more favorable than neoadjuvant TIP-based chemotherapy, as patients in the former group showed relatively better clinical outcomes. However, evidence in this setting is scarce and limited to small, retrospective studies.

The two most common regimens include the combination of paclitaxel, ifosfamide, and cisplatin (TIP) or 5-FU with cisplatin and docetaxel (TPF) [15]. These combination regimens offer a modest efficacy, with a complete response rate of 15% and an objective response rate of 50%. However, survival outcomes are suboptimal, with a 5-year OS rate of <50% in patients with clinical N1-3 disease [33,34]. Although lymph node dissection can benefit this group of patients, only 16–20% will be cured [33,34]. Rose et al. investigated the efficacy and safety of neoadjuvant chemotherapy in patients with locally advanced disease and clinical node metastasis. Results showed a median OS of 37 months (95% CI 23.8–50.1) and PFS of 26 months (95% CI 11.7–40.2). This study demonstrated how neoadjuvant chemotherapy with lymphadenectomy is well tolerated and can reduce disease burden in patients with locally advanced penile cancer [35]. However, most patients in this cohort had cT3-4 disease. Another multicenter retrospective study showed improved outcomes for patients with PSCC who received adjuvant chemotherapy and who had positive lymph nodes after surgery. Specifically, the estimated OS was 21.7 months (IQR: 11.8–104) in patients who received adjuvant chemotherapy compared to 10.1 months (IQR: 5.6–48.1) in those who did not (*p* = 0.048). The benefit of adjuvant chemotherapy was independently associated with improved OS on multivariate analysis (HR 0.40; 95% CI: 0.19–0.87; *p* = 0.021) [19]. As most patients had bilateral lymph node involvement, this study further supports the notion that bulky bilateral node disease may derive the highest clinical benefit from perioperative chemotherapy, albeit the results require further validation. The international penile advanced cancer trial (InPACT, NCT02305654) is currently evaluating the optimal role of perioperative chemotherapy and radiotherapy in patients with PSCC extent of regional lymph node dissection [36]. This phase III trial, with over three-quarters of patients accrued, is expected to be completed by 2027.

## 5. Metastatic Disease

### 5.1. Chemoimmunotherapy

Chemoimmunotherapy has emerged as a promising approach to enhance response rates and survival outcomes in advanced PSCC by leveraging the immunogenic potential of cytotoxic agents alongside PD-1/PD-L1 blockade. One of the first prospective studies in this space, the HERCULES trial (LACOG 0218, NCT04224740), was a phase II, single-arm trial evaluating the combination of pembrolizumab and platinum-based chemotherapy as first-line treatment. Thirty-seven patients were enrolled across 11 Brazilian centers, with 33 patients evaluable for efficacy. The confirmed objective response rate (ORR) was 39.4% (95% CI, 22.9–57.9), including one complete response and 12 partial responses. Tumor shrinkage was observed in 75.8% of patients. Median progression-free survival (PFS) and OS were 5.4 and 9.6 months, respectively. Immune-related adverse events occurred in 21.6% of patients, with 5.4% showing grade 3–4 adverse events, and 10.8% discontinued treatment due to toxicity. Notably, ten patients experienced grade 5 events, none of which were attributed to the study treatment. Exploratory biomarker analysis showed higher response rates among patients with high tumoral mutation burden (TMB) and HPV16-positive tumors. Specifically, patients with a higher TMB status exhibited a higher response rate compared to patients with low TMB status (75% in high vs. 36.4% in low). A similar trend was seen in patients with HPV16-positive status (HPV16-positive 55.6% vs. HPV16-negative 35.0%) [37].

Following the rationale of combining immune checkpoint blockade with cytotoxic chemotherapy, the EPIC-A trial (NCT95561634) investigated cemiplimab (anti-PD-1) in combination with standard-of-care platinum-based chemotherapy followed by cemiplimab maintenance. This open-label, multicenter phase II study enrolled 29 patients with locally advanced or metastatic PSCC [38]. Patients received four cycles of cisplatin-based chemotherapy (either cisplatin + 5-FU or TIP) alongside cemiplimab 350 mg IV every three weeks, followed by cemiplimab monotherapy for up to two years. At 12 weeks, the clinical benefit rate (CBR) was 62%, with partial responses observed in 52% of patients and stable disease in 10.3%. One complete response was noted at 21 weeks. Median PFS was 6.2 months, and median OS was 15.5 months. A total of seven patients discontinued treatment due to an adverse event, four of which were related to cemiplimab (14%). These findings further support the role of first-line chemoimmunotherapy in improving outcomes in PSCC [38]. A recent phase 2 trial, NCT04475016, evaluated the efficacy of toripalimab, nimotuzumab, and taxol-based chemotherapy (TNT) followed by consolidative surgery in patients with locally advanced PSCC. The objective response rate (ORR) was 82.8% (95% CI 64.2–94.2), with a 2-year OS and PFS rates of 72.4% and 65.5%, respectively. There were no treatment-related deaths. This study underscores the promising anti-tumor activity of neoadjuvant TNT, which could inform future clinical trial design [39].

### 5.2. Immunotherapy as Monotherapy

Immunotherapy as monotherapy has demonstrated modest activity in patients with advanced PSCC, particularly those who are ineligible for or have progressed to conventional chemotherapy regimens. The AcSé trial (NCT03012581), a phase II study, evaluated nivolumab monotherapy in 43 patients with previously treated advanced PSCC. The ORR at 12 weeks was 14%, with four partial responses and stable disease in 32% of patients [40]. Median PFS was 2.9 months and OS was 8.5 months, reflecting limited benefit from PD-1 blockade monotherapy in this unselected, heavily pretreated population [40]. Similarly, the ORPHEUS trial (NCT04231981) assessed retifanlimab, an anti–PD–1 antibody, in 18 patients with advanced PSCC [41]. The study reported an ORR of 16.7% (95% CI 5.8–39.2), with three partial responses and a disease control rate of 22.2%. The CBR was 22.2% (95% CI 6.4–47.6). Median PFS was 2.0 months (95% CI 1.6–3.3) and median OS was 7.2 months (95% CI 3.0–9.8). These results highlight the signals of clinical activity in advanced/metastatic PSCC, with no concerning safety signals. However, its most substantial limitation is the small sample size [41].

These results are consistent with the modest activity of immunotherapy as monotherapy observed in other squamous cell carcinomas of the anogenital tract. The PERICLES trial (NCT03686332), a nonrandomized, single-center phase II study, evaluated atezolizumab (anti-PD-L1) with or without concurrent radiotherapy in 32 patients with unresectable advanced PSCC [42]. Although the trial did not meet its primary endpoint, showing a 1-year PFS of 12.5% (95% CI, 5.0–31.3), some durable responses were observed. Specifically, the response rate was 16.7%, comprising two complete and three partial responses. Median OS was 11.3 months. However, radiotherapy-related grade 3–4 toxicities were reported in 65% of patients, and immune-related adverse events occurred in 62.5%. Exploratory biomarker analyses revealed improved PFS in patients with high-risk HPV–positive tumors and elevated intratumoral CD3+ CD8+ T-cell infiltration, highlighting potential predictors of response to ICI monotherapy and the utility of biomarker-selected populations in clinical trial design [42]. The EPIC-B trial (NCT95561634) expanded on these findings by evaluating the efficacy and safety of cemiplimab as a first-line treatment in patients with locally advanced or metastatic penile cancer [43]. This was a single-arm, multi-center phase II study where patients received cemiplimab 350 mg IV D1 every 3 weeks (Q3W) for up to 34 cycles. At 12 weeks, the CBR was 38.9% (95% CI, 20.3–61.4), and the ORR was 16.6% (95% CI, 5.8–39.2), with three patients showing partial responses and four experiencing stable disease. Notably, throughout treatment, 27.7% of patients responded to treatment, including one complete response and four instances of stable disease. Median PFS was 2.3 months (95% CI 1.0–3.8), and median OS was 6.8 months (95% CI 3.7–9.8). The safety profile showed considerable toxicity, with 31% of patients showing any grade adverse events related to cemiplimab and 26% showing ≥grade 3 adverse events. There were two instances of grade 5 adverse events, one of which (toxic epidermal necrolysis) was related to treatment [43].

Similarly, a separate basket trial (NCT02721732) evaluated pembrolizumab in patients with rare cancers whose tumors had progressed to standard therapies. Three patients with penile cancer who had progressed to platinum-based chemotherapy were included. Two patients experienced early progression, while one patient with a microsatellite instability-high (MSI-H) tumor achieved a partial response [44]. This finding highlights the potential relevance of tumor-agnostic biomarkers, such as MSI-H, in identifying patients who are more likely to benefit from immunotherapy and those who may derive limited therapeutic benefit from pretreatment in unselected patients. The ALPACA trial (NCT03391479) is evaluating avelumab as monotherapy in patients with locally advanced or metastatic penile cancer who are ineligible for, or have progressed on, platinum-based chemotherapy [45]. While results are pending, the trial reflects the growing interest in defining the role of ICI monotherapy in biomarker-enriched populations.

### 5.3. Immunotherapy-Based Combinations and Tyrosine Kinase Inhibitors

Additional insights have emerged from basket trials that included patients with PSCC, although data remain limited. The BUTCVH trial (NCT03333616), a phase II study evaluating the combination of nivolumab and ipilimumab, enrolled five patients with advanced PSCC. No objective responses were observed, with two patients achieving stable disease and three experiencing disease progression, indicating minimal activity of dual checkpoint inhibition in this setting [46].

Similarly, the LATENT trial (NCT03357757) is a phase II non-randomized clinical trial evaluating avelumab (anti-PD-L1) in combination with valproic acid for virus-associated malignancies, including p16-positive PSCC. Results are eagerly awaited. Notably, this study highlights the therapeutic use of valproic acid in viral-related malignancies. The rationale for its use is based on its inhibitory effect on histone deacetylases, which play a crucial role in chromatin remodeling. This prevents cellular division and induces apoptosis. By combining valproic acid with avelumab, investigators seek to enhance the immunologic response against viral-infected tumoral cells [47]. Interestingly, a phase 2 study (NCT05526989) is currently evaluating the efficacy and safety of niraparib and dostarlimab in patients with advanced refractory PSCC. This is an exciting trial as it incorporates immunotherapy and PARP inhibitors as dual therapy for patients with PSCC. This study is actively recruiting [48].

The combination of immunotherapy with checkpoint inhibitors is also being analyzed. The ICONIC trial (NCT03866382) seeks to assess the efficacy of nivolumab, ipilimumab, and cabozantinib in rare genitourinary tumors. Among nine patients with metastatic PSCC treated to date, four (44%) achieved partial responses, suggesting promising activity. Additionally, a separate phase II trial (NCT04475016) is testing triprilimab (anti–PD–1) combined with nimotuzumab (anti-EGFR) and TIP chemotherapy in patients with locally advanced PSCC. Among the 29 enrolled patients, 24 (82.8%) underwent consolidative surgery, with 14 (48.3%, 95% CI, 29.4–67.5) achieving a pathologic complete response. The ORR was 82.8% (95% CI, 64.2–94.2), with an estimated 2-year PFS rate of 65.5% (95% CI, 48.3–82.7), and a 2-year OS of 72.4% (95% CI, 56.1–88.7). Notably, biomarker analysis showed that responders presented higher PD-L1 expression proportion scores than non-responders (*p* = 0.019). Likewise, PFS (*p* = 0.006) and OS (*p* = 0.045) benefit was observed in patients with PD-L1 positive expression [39]. Another phase II study (NCT01728233) is evaluating dacomitinib, an irreversible pan-epidermal growth factor receptor (HER) inhibitor in patients with advanced PSCC (N2-3 or M1). A total of 28 patients were treated, of which one showed a complete response and eight showed partial responses (ORR 32.1%). The 12-month PFS was 26.2% (95% CI 13.2–51.9) and the 12-month OS was 54.9% (95% CI 36.4–82.8). Overall, dacomitinib was active and well tolerated in patients with advanced PSCC and can be a promising therapeutic option [49].

### 5.4. Oncolytic Vaccines

Because HPV-related penile cancer is prone to immunomodulation by targeting the HPV-driven oncogenic proteins, various researchers are evaluating the efficacy of therapeutic HPV vaccines. These vaccines stimulate the immune system by presenting mutated versions of the E6 and E7 proteins and enhancing the CD8+ cytotoxic T-cell response [50]. Although no trials evaluating these vaccines in the context of penile cancer have been published, some ongoing trials are concurrently evaluating HPV-targeting therapeutic vaccines and ICI. The design of these trials is based on the notion that dual immunologic modulation may prove superior to either agent alone by enhancing the immunologic response and mitigating tumor-mediated immunosuppression. For example, a phase I/II trial (NCT04432597) evaluated the HPV vaccine PRGN-2009 alone or in combination with bintrafusp alfa, a bifunctional TGF-β “trap”/anti-PD-L1 fusion protein in patients with HPV-associated cancers. Bintrafusp works by simultaneously blocking the TGF-β and PD-L1 pathway, leading to an increased antitumor immune response. Results showed an ORR of 30.0% (95% CI 6.7–65.3). The median OS was 7.4 months (95% CI, 2.9–26.8) for the dose-escalation cohort and 12.5 months (95% CI, 9.6-inestimable) for the combination cohort. Post vaccination, 88% of patients (14/16) developed T-cell responses to HPV-16 and/or HPV-18 [51]. This study showed a high tolerability profile and a high level of HPV-specific T-cell response. A similar study (NCT04287868) is evaluating the clinical activity of the HPV-16 therapeutic vaccine PDS0101, PDS01ADC (an anti-interleukin 12 ADC), and bintrafusp alfa in advanced HPV-associated cancers. A total of 50 eligible patients were included, with an ORR of 35.7% (95% CI, 12.8–64.9%), and a median OS of 42.4 months (95% CI, 8.3 months-not estimable). Among patients with HPV-16-positive tumors (74%), the ORR was 62.5% (95% CI, 24.5–91.5%). Overall, this combination showed promising antitumor activity and improved OS in patients with HPV-16-positive cancers [52]. Another phase II trials (NCT02426892) is evaluating the efficacy of nivolumab and ISA 101, a synthetic long-peptide HPV-16 vaccine inducing HPV-specific T cells, in patients with incurable HPV-16-positive cancers. This trial included 24 patients, with an ORR of 33% (90% CI, 19–50%), a median duration of response of 10.3 months (95% CI, 10.3 months-inestimable), and a median PFS of 2.7 months (95% CI, 2.5–9.4 months) [53]. Although these trials show promising efficacy, it’s important to note they were conducted in the context of basket trials including HPV-associated cancers, and not in penile cancer-specific cohorts. This underscores the need for dedicated studies in this disease setting.

### 5.5. Antibody–Drug Conjugates

Antibody–drug conjugates (ADCs) are formed by a monoclonal antibody, a linker, and a cytotoxic payload. The monoclonal antibody targets a specific antigen and, when bound, delivers the cytotoxic payload. Most payloads are composed of DNA synthesis inhibitors (calicheamicins, duocarmycins, pyrrolobenzodiazepines), microtubule-targeting agents (auristatins, maytansines), and topoisomerase-1 inhibitors (SN-38, exatecan, irinotecan, CPT-11, belotecan) [54]. Human epidermal growth factor receptor 2 (HER-2) ADCs are considered low-toxicity agents and have shown improved efficacy for several advanced cancers, including breast, lung, and bladder. HER-2 is expressed in approximately 25% of PSCC samples, and its expression has been associated with worse prognosis, higher tumoral stages, and poor overall survival [55]. Weiten et al. evaluated the expression of trophoblast cell surface antigen-2 (TROP-2) in localized and metastatic PSCC, analyzing the degree of immunohistochemical (IHC) expression in primary and patient-matched metastatic tissue. Their results showed an increase in TROP-2 expression in individuals with PSCC compared to healthy controls (*p* < 0.0001), with a notable elevation in TROP-2 expression in serum samples obtained from patients with PSCC (*p* < 0.01). This strong antigenic expression suggests that TROP-2-targeting ADCs (sacituzumab govitecan or datopotamab deruxtecan) could be explored in this disease space [56]. A phase II clinical trial (SMART/NCT06161532) is currently evaluating SG with or without atezolizumab in rare genitourinary tumors, including penile cancer. This study is actively recruiting patients [57].

Nectin-4, a member of the adherens intercellular junction and a ligand for the immune checkpoint inhibition family, is expressed in approximately 89% of PSCC samples [58]. Case reports have described the efficacy of enfortumab vedotin (EV) for treating PSCC. In their findings, Fahey et al. describe how a heavily pretreated patient with cisplatin-based chemotherapy showed radiographic and clinical response after two treatment cycles with EV [59]. A phase II clinical trial (NCT06104618) is evaluating EV in patients with PSCC and regional lymph node involvement or distant metastatic (M1) disease. This study is actively recruiting patients [60]. Given the limited durable response observed with monotherapy, there is a pressing need to develop combination strategies. Pairing ADCs with immunotherapy or tyrosine kinase inhibitors could be the next step in therapeutic trial design for PSCC. This could potentially improve treatment efficacy and response rate beyond those achieved by current treatment regimens. However, these strategies remain in the early phases of clinical investigation, and their safety, efficacy, and long-term benefit in penile cancer will need to be confirmed in larger, disease-specific trials before they can be incorporated into standard practice. A description of select clinical trials evaluating different modalities of systemic therapy is shown in Table 1. A visual representation of the various treatment modalities for penile cancer is shown in Figure 1.

## 6. Immune Implications and Microenvironment in Penile Cancer

During an immune response, T-cell activation leads to upregulation of PD-1, which binds to PD-L1 on the surface of antigen-presenting cells to modulate the effector response of T-cells, a process known as “immunosuppression” [61,62]. Cancer models have shown an increased expression of PD-L1 as an escape mechanism to bypass the effector immunologic reaction by inhibiting T cells, leading to unchecked growth and progression. In PSCC, up to 40–69% of tumors will express PD-L1 [63,64]. Since a significant proportion of PSCC cases are HPV-driven, modulating the immune system has proven to be a plausible mechanism for disease control. Characterizing the tumor microenvironment is of utmost importance for understanding possible therapeutic targets. Both the innate and adaptive immune systems play a role in tumorigenesis. For example, CD8+ T cytotoxic cells, CD4+ T helper cells, and natural killer cells play a role in antitumor activity. Conversely, T regulatory cells, tumor-associated macrophages (TAMs), and myeloid-derived suppressor cells (MDSCs) hinder the antitumor response [65,66]. A study evaluating the presence of CD8+ cytotoxic T cells, FOXP3+ T-regulatory cells, and TAMs in 213 patients with PC demonstrated that a low stromal CD8+ cytotoxic T cell count was significantly associated with lymph node metastasis (OR 0.60, CI 0.37–0.98; *p*  =  0.04) [67]. Subgroup analysis of tumoral HPV-negative and high-risk HPV-positive specimens showed a positive correlation between increased tumor grade and degree of infiltration of CD14+, CD68+, and CD163+ cells in the intratumoral and peritumoral compartments, suggestive of an avid macrophage-driven response [67]. A similar study explored the immune microenvironment in 57 men with PSCC across different tumor stages to identify key immune cell markers implicated in tumorogenesis. Authors reported an increase in the density of immune effector cells in patients with node-negative disease and a rise in inhibitory immune mediators like type 2 macrophages. Patients with N1 and N2-3 disease showed an upregulation of PD-L1. Interestingly, there were no differences in immune cell densities with HPV status [68]. Conversely, Miyagi et al. describe a difference between HPV-negative and HPV-positive tumors. Their results suggest that HPV-negative tumors display low immune cell infiltration in the tumor microenvironment, while HPV-positive tumors elicit an immune-exhausted phenotype characterized by terminally exhausted CD8+ T cells, M2-like macrophages, and hypoxic signatures [69].

Similarly, the PI3K/mTOR/AKT pathway might represent an alternative option for the treatment of penile cancer. A study performed by Thomas et al. evaluated the molecular profiling of primary advanced penile cancer tumor tissue in 76 patients. Their results showed that a higher expression of AKT was significantly more prevalent in high-grade tumors and more predictive of disease-specific survival and overall survival (all, *p* < 0.05). Moreover, higher levels of p4EBP1 were also associated with an inferior disease-specific survival (*p* = 0.03). This preliminary study shows the implication of mTOR/AKT signaling pathway-related proteins in clinical outcomes among patients with advanced penile cancer [70]. A similar analysis of 94 patients with penile cancer showed that an increased expression of the receptor tyrosine kinase cellular mesenchymal–epithelial transcription factor (c-MET) and its associated signaling protein PPARg were predictive of inferior disease-specific survival, further elucidating the role of molecular markers as surrogates for disease aggressiveness [71].

Altogether, the immunogenic profile of PSCC, characterized by frequent PD-L1 expression and immune cell infiltration, supports the rationale for immune checkpoint blockade. In addition, the implication of PI3K/mTOR/AKT pathway-related proteins in disease outcomes is expanding the potential roster of therapeutic targets. These features, particularly in HPV-associated tumors, suggest that immune evasion plays a key role in disease progression.

## 7. Conclusions

Penile cancer is a rare but aggressive malignancy with a disproportionately high burden in resource-limited settings. While organ-preserving strategies and risk-adapted lymphadenectomy have improved outcomes in localized disease, patients with advanced or metastatic PSCC continue to face poor prognoses, underscoring the urgent need for more effective systemic therapies. Recent advances in immunotherapy have shown preliminary signs of clinical activity and highlight the immunogenic nature of PSCC. Trials such as HERCULES, InPACT, ALPACA, and others underscore ongoing efforts to help clarify the role of these agents in both treatment-naïve and pretreated populations. Ultimately, improving survival in advanced PSCC will require biomarker-driven strategies, broader access to clinical trials, and collaborative efforts to tailor therapies to this understudied malignancy’s unique molecular and immunological landscape.

## Figures and Tables

**Figure 1 cancers-17-02956-f001:**
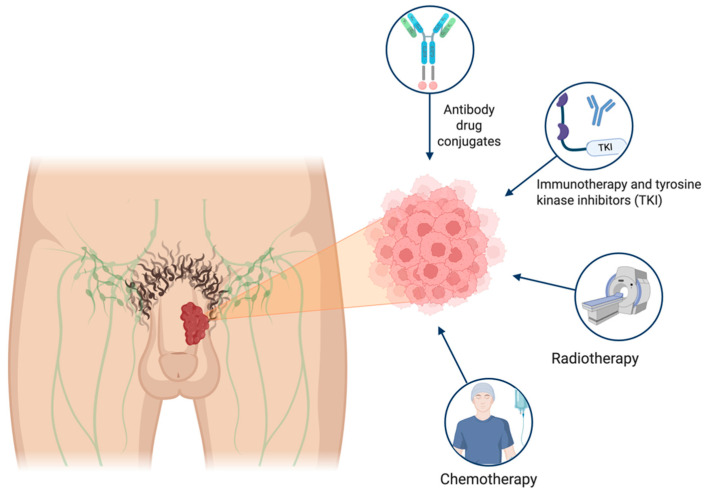
Treatment landscape for penile cancer. Patient with penile mass and focus on cancer cells and emerging therapeutic options.

**Table 1 cancers-17-02956-t001:** Select clinical trials evaluating systemic therapy options for penile cancer.

Study	Regimen/Intervention	Design	Patient Population	Response	Toxicity
HERCULES (NCT04224740)	Pembrolizumab + platinum-based chemotherapy	Phase II, single-arm	Advanced/metastatic PSCC, treatment-naïve	ORR 39.4%, median PFS 5.4 mo, OS 9.6 mo	21.6% AEs (5.4% grade 3–4), 10.8% discontinuation
EPIC-A (NCT95561634)	Cemiplimab + chemotherapy followed by maintenance cemiplimab	Phase II, non-randomized	Locally advanced/metastatic PSCC, treatment-naïve	CBR 62%, median PFS 6.2 mo, OS 15.5 mo	23% cemiplimab-related, 31% chemo-related, 14% discontinued due to cemiplimab
EPIC-B (NCT95561634)	Cemiplimab monotherapy	Phase II, single-arm	Locally advanced/metastatic PSCC, chemo-ineligible	CBR 38.9%, ORR 16.6%, median PFS 2.4 mo, OS 10.7 mo	31% any grade, 1 death
PERICLES (NCT03686332)	Atezolizumab ± radiotherapy	Phase II, non-randomized	Unresectable advanced PSCC	ORR 16.7%, median OS 11.3 mo	62.5% AEs
AcSé (NCT03012581)	Nivolumab monotherapy	Phase II, multi-cohort	Advanced PSCC, post-chemotherapy	ORR 14%, PFS 2.9 mo, OS 8.5 mo	14.3% grade 3–4 AEs
ORPHEUS (NCT04231981)	Retifanlimab monotherapy	Phase II	Advanced PSCC	ORR 16.7%, PFS 2.0 mo, OS 7.2 mo	Not reported
BUTCVH (NCT03333616)	Nivolumab + ipilimumab	Phase II basket	Advanced PSCC	0% ORR, 2 SD, 3 PD	Not reported
NCT02721732	Pembrolizumab	Phase II basket	PSCC post platinum-based chemotherapy	1 PR (MSI-H), 2 PD	Not reported
ALPACA (NCT03391479)	Avelumab monotherapy	Phase II (ongoing)	Locally advanced/metastatic PSCC, post-chemotherapy or ineligible	Ongoing	Ongoing
LATENT (NCT03357757)	Avelumab + valproic acid	Phase II (ongoing)	p16+ virus-associated cancers incl. PSCC	Ongoing	Ongoing
NCT03866382	Nivolumab + ipilimumab + cabozantinib	Phase II (ongoing)	Rare GU cancers incl. PSCC	ORR 44% (4/9 pts)	Ongoing
NCT04475016	Triprilimab + nimotuzumab + TIP	Phase II (ongoing)	Locally advanced PSCC	Ongoing	Ongoing
NCT01728233	Dacomitinib	Phase II	Locally advanced PSCC	ORR 32.1%, 1 CR, 8 PR	10.7% grade 3 ≥ AEs.
NCT06104618	Enfortumab vedotin	Phase II	Metastatic or Unresectable PSCC	Ongoing	Ongoing
NCT06161532	Sacituzumab govitecan +/- atezolizumab	Phase II	Rare GU tumors, including penile cancer	Ongoing	Ongoing
NCT04287868	PDS0101, PDS01ADC, and bintrafusp alfa	Phase I/II	HPV-associated cancers	ORR 35.7% and median OS 42.4	Grade 3 and 4 AEs occurred in 52%.
NCT02426892	Nivolumab + ISA 101	Phase II	HPV-16-positive cancers	ORR 33%, median PFS 2.7 months, median OS 17.5 months	Grades 3 to 4 AEs occurred in 2 patients.
NCT04475016	Toripalimab + nimotuzumab + taxol-based chemotherapy	Phase II	Locally advanced PSCC	ORR 82.8%, 2-year OS 72.4% and 2-year PFS 65.5%.	Grade 3–4 AEs occurred in 41.4%.
NCT05526989	Niraparib + dostarlimab	Phase II	Refractory PSCC	Ongoing	Ongoing

AEs, adverse events; CBR, clinical benefit rate; GU, genitourinary; HPV, human papilloma virus; ORR, objective response rate; OS, overall survival; PFS, progression-free survival; PD, progressive disease; PR, partial response; PSCC, penile squamous cell carcinoma; TIP, paclitaxel, ifosfamide, and cisplatin.

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
