# Peer review of "Evaluating the Evolving Treatment Landscape of Systemic Therapies in Penile Cancer"

_cancers, 2025, doi:10.3390/cancers17182956_

Round 1
Reviewer 1 Report
Comments and Suggestions for Authors
This is a deep narrative review from experts in the field about a poor studied cancer, so it is interesting to have it published. However, there are some issues that should be addressed.
All the references should be revised as there are several concerns. For example; reference 24 in cited in a context of recurrence, but it is about laser topical treatments. Please review and update references such us 21 with new v2. 2025 NCCN guidelines. Or Reference 30 from 2004… to the most updated 2025 EAU guideline. There are several papers from 2004 and 2014 until now showing evidence (despite being non-randomised) about the role of QT and RT in this setting. Reference 32 is duplicated. 32 and 34 is the same.
Was there any research protocol, terms, papers selection criteria or similar?
The sections distributions could be improved. The authors structured the review from less to more advance disease but the title is about systemic therapy which until now is useless in localized N0 disease. In the middle the include a section of immune environment and implications. Please reorganized the structure in a more comprehensive way.
Palpable inguinal Node disease section should be revised. Some statements “The National Comprehensive Cancer Network (NCCN) and European Association of Urology (EAU) guidelines recommend neoadjuvant chemotherapy, typically the TIP regimen (paclitaxel, ifosfamide, cisplatin), followed by consolidative radical surgery and inguinal lymph node dissection in those who respond to the treatment, as upfront surgery alone has a low probability of cure.” are not clearly supported. It should be specified that the recommendation is weak and in cases of bulky or bilateral disease. In this scenario as the evidence is sparse and low quality in most of the cases, the statements should be in line with this. This issue is also notice in the neo-adjuvant QT section. Neoadjuvant section in N1-2 (no bulky) disease is not so clearly demonstrated as stated in the review and the references used are not focused in this state.
In the section 6.2 add more information in the title to clarify it is just immunotherapy in monotherapy as the previous section also include immunotherapy trials.
Please take dacomitinib out from immunotherapy section as it is not a specific immunomodulator treatment. Also, double check the last paragraph trials, there are some of them studying combinations treatments, so they suit more in the previous section Chemoimmunotherapy.
Author Response
"Please see the attachment."

Reviewer 2 Report
Comments and Suggestions for Authors
With great interest, I reviewed the manuscript evaluating systemic therapy strategies in penile cancer, with a focus on evolving evidence and emerging treatment modalities. This is a clinically important and rare disease context, where robust data are scarce and guidance often relies on small prospective trials, retrospective analyses, or extrapolation from other squamous cell carcinomas. The topic is relevant to oncologists and urologists managing advanced penile cancer. However, several important points limit the current value and novelty of the review, and substantial revisions are necessary before the manuscript can be considered for publication.
Major comments
Novelty and scope: While the manuscript provides a general overview of systemic treatment options, much of the content repeats data and perspectives already presented in recent reviews (e.g., Hakenberg et al., Eur Urol Oncol2023; Ottenhof et al., Nat Rev Urol 2022). The authors should clearly articulate what is new in their synthesis — for example, unique insights, comparative interpretation, or updated trial data published in 2023–2024.
The section on immune checkpoint inhibitors is largely descriptive and does not critically assess predictive biomarkers, optimal sequencing, or combination strategies, despite several recent case series and early-phase trials in penile cancer and related SCCs.
Incomplete literature coverage: Important recent studies are missing, including updates from basket trials (e.g., KEYNOTE-158 penile cancer subset, CheckMate-358 SCC cohorts), and ongoing trials combining chemotherapy with immunotherapy or targeted agents.
The manuscript does not sufficiently address the role of molecular profiling and actionable mutations (e.g., EGFR alterations, PI3K pathway) in guiding systemic therapy, which is increasingly discussed in the literature.
Critical appraisal: The review reads as a narrative summary rather than a critical synthesis. Statements regarding efficacy of cisplatin-based combinations, taxanes, or targeted agents are not consistently supported by comparative outcomes or statistical context.
The authors should clearly indicate the level of evidence (e.g., phase, sample size, outcome measures) when discussing each regimen. Without this, readers cannot judge the strength of the recommendations.
Structure and clinical relevance: There is limited discussion of practical decision-making — for example, how to select between TIP (paclitaxel–ifosfamide–cisplatin) and other cisplatin-based regimens in metastatic disease, or how to approach frail patients.
A structured treatment algorithm summarizing current practice and investigational options would add significant value for clinicians.
Regional disparities and access to care: The authors should briefly address how systemic therapy choices may differ between high-income and low/middle-income countries, as access to drugs and supportive care impacts real-world applicability of guidelines.
Minor Comments
Tables and figures: Consider adding a summary table of key systemic therapy trials in penile cancer, including design, patient numbers, response rates, PFS, and OS.
Any figures illustrating evolving treatment paradigms should be self-explanatory with clear legends.
Language and consistency: Minor grammatical corrections are needed. Ensure consistent use of terms such as “systemic therapy,” “immune checkpoint inhibitor,” and “targeted therapy.”
References: Verify that all citations are up-to-date and representative. Several relevant references from 2023–2024 are missing.
Author Response
"Please see the attachment."

Reviewer 3 Report
Comments and Suggestions for Authors
I read with great interest these review.
Below are my main suggestions to improve the quality of the review.
A brief section on the methods and on the screening of the article is needed. Did authors include all RCTs for the various stages of penile cancer? How was the selection made? Is this a systematic or a narrative review?
I believe some references should be added with more updated papers.
Line. 59- 60. DOI: 10.3390/curroncol30120765
Line 110-114 DOI: 10.1200/PO-25-00045
looking forward for the revised version.
Author Response
"Please see the attachment."

Round 2
Reviewer 1 Report
Comments and Suggestions for Authors
all comments have been addressed
Reviewer 2 Report
Comments and Suggestions for Authors
No further comments - revised version improved quality of the manuscript